# Effects of Prolactin Inhibition on Lipid Metabolism in Goats

**DOI:** 10.3390/ani14233364

**Published:** 2024-11-22

**Authors:** Xiaona Liu, Chunhui Duan, Xuejiao Yin, Xianglong Li, Meijing Chen, Jiaxin Chen, Wen Zhao, Lechao Zhang, Yueqin Liu, Yingjie Zhang

**Affiliations:** 1College of Animal Science and Technology, Hebei Agricultural University, Baoding 071001, China; liuxiaonahau@163.com (X.L.); duanchh211@126.com (C.D.); chenmeijing815@126.com (M.C.); chenjiaxin1226@163.com (J.C.); zhaowen970920@163.com (W.Z.); 18531132767@163.com (L.Z.); 2College of Animal Science and Technology, Hebei Normal University of Science & Technology, Qinhuangdao 066004, China; bdyinxuejiao@foxmail.com (X.Y.); 15203358192@163.com (X.L.)

**Keywords:** goats, prolactin, lipid metabolism, liver, adipose tissue

## Abstract

The effects of prolactin on lipid metabolism are unclear. Therefore, we studied the effect of prolactin on lipid metabolism in cashmere goats by using the prolactin inhibitor bromocriptine. The results showed that prolactin inhibition reduced serum hormone-sensitive lipase levels but had inconclusive effects on body weight and average daily feed intake. Hematoxylin–eosin staining showed that prolactin inhibition did not cause obvious pathological changes in the liver, subcutaneous adipose, or perirenal adipose tissues. However, the inhibition of prolactin decreased the area of perirenal adipocytes. In the liver, prolactin inhibition increased the expression of prolactin, long-form, and short-form prolactin receptor genes, and lipogenesis and lipolysis genes. In subcutaneous adipose tissue, prolactin inhibition increased the expression of the short-form prolactin receptor gene and decreased the expression of some lipogenesis and lipolysis genes. In perirenal adipose tissue, the inhibition of prolactin decreased the expression of prolactin, sterol regulatory element binding transcription factor 2, and 3-hydroxy-3-methylglutaryl-CoA reductase genes. In conclusion, the effects of prolactin on lipid metabolism are different in different goat tissues.

## 1. Introduction

Prolactin (PRL), a single-chain peptide hormone, is mainly secreted by the anterior pituitary gland and belongs to the PRL/growth hormone family [1,2]. In addition to its classic effects on lactation and reproduction, increasingly more evidence has recently proven that PRL plays an important role in metabolic homeostasis [3,4]. The role of PRL in metabolism has been reported to be closely related to its serum concentration [5,6,7]. Studies show that high levels of PRL promote weight gain, obesity, metabolic syndrome, and impairment of glucose–insulin and lipid profiles [8,9,10,11]. Recent studies have found that low PRL levels have also been shown to adversely affect weight gain, glucose, and lipid metabolism, resulting in an increased prevalence of metabolic syndrome [12,13,14]. These results suggest that abnormal PRL levels may lead to metabolic homeostasis disorder. In ruminants, PRL receptors (PRLRs) come in two distinct forms, the long-form PRLR (LPRLR) and the short-form PRLR (SPRLR) [15,16]. Although both LPRLR and SPRLR are active upon binding to PRL and can activate downstream signals, differences can be seen in the signaling pathways of these two receptors [17]. PRLR is widely distributed throughout the body [2,4,18], and an increasing number of studies have demonstrated that PRLR is highly expressed in the liver [4,19,20]. The liver is a central organ for lipid metabolism and plays an important role in cellular metabolic processes, such as lipid digestion, absorption, transport, and catabolism [21], which may imply that PRL plays an important metabolic role in the liver.

PRL is produced not only by the pituitary gland but also by adipose tissue [22,23,24]. This finding was discovered in a study that used human adipose tissue as a negative control for PRL release [25], and recent studies have overturned the previous assumption that PRLR is not expressed in adipose tissue [4,26]. Therefore, the study of PRL in adipose tissue has attracted increasingly more attention. During the differentiation of rat epididymal preadipocytes, the expression of *LPRLR* and *SPRLR* increases manifold [27]. In human mammary preadipocytes, PRLR shows an initial decrease, followed by an increase during adipogenesis [28]. PRLR deficiency results in a reduction in the size of the fat depot, which is due to a decrease in the number of adipocytes rather than a change in their volume [29]. In goats, LPRLR is expressed in subcutaneous adipose tissue but at slightly lower levels than in their livers [19]. PRL also plays a role in the differentiation or trans-differentiation of brown adipocytes [30]. Doknic et al. demonstrated that excess PRL promotes lipogenesis [31]. In women who are obese, PRL promotes visceral fat storage [32]. Moreover, PRL inhibits lipid storage and adipokine release in adipose tissue [6]. PRL deficiency also has deleterious effects on the function of adipose tissue in women [12]. PRL can inhibit the mRNA and protein expression of fatty acid synthase (FAS) in adipocytes [33]. In the liver, PRL ameliorates hepatic steatosis via the CD36 pathway [34]. PRL has been shown to inhibit isoproterenol-stimulated lipolysis in rat and human adipose tissues [27] but has no effect on lipolysis in mouse adipose explants [35]. This indicates that the complex regulation mechanism of PRL on lipid metabolism needs to be further elucidated.

Adipose tissue is generally categorized into subcutaneous and visceral adipose tissues, and the two types of adipose tissue have distinct differences, with visceral adipose tissue possessing a higher lipid turnover rate than subcutaneous adipose tissue [36]. The simultaneous collection of two different types of adipose tissue is more conducive to exploring the role of PRL in lipid metabolism. Additionally, bromocriptine (BCR), a dopamine agonist that inhibits PRL secretion, has been widely used to study the function of PRL [37,38,39,40]. Therefore, our study aimed to elucidate the effects of PRL on lipid metabolism in cashmere goats. The potential mechanism of PRL regulating lipid metabolism in cashmere goats was investigated by observing the changes in blood indexes, section morphology, and gene expression in liver, subcutaneous adipose, and perirenal adipose tissues after the inhibition of PRL.

## 2. Materials and Methods

### 2.1. Animals and Management

This experiment was conducted from September to October 2020 at a cashmere goat farm in Qinglong, Qinhuangdao City, China (longitude 118°95′ E, latitude 40°41′ N, 245 m above sea level). The goats were dewormed with ivermectin before the start of the experiment. Twenty goats were housed in twenty individual pens. During the experimental period, all goats had free access to fresh water, and feeding and management were synchronized with the goat farm. The goats were fed twice a day. The diet was composed of 13.5% protein concentrate (purchased from Chaoyang Tianqin Feed Company, Chaoyang, China), 37.8% corn, and 48.7% corn stalk. The protein concentrate included crude protein 43.8%, crude ash 24.7%, crude fiber 12.0%, calcium 1.9%, sodium chloride 2.4%, lysine 1.2%, total phosphorus 1.0%, and moisture 13.0%. The dietary metabolizable energy rate was 8.5 MJ/kg, and the crude protein level was 9.5% [41].

### 2.2. Experimental Design and Sample Collection

Twenty eleven-month-old male Yanshan cashmere goats were randomly assigned to two groups (*n* = 10) using a completely randomized design: a control (CON) group and a bromocriptine-treated (BCR) group. The goats in the BCR group were treated with bromocriptine (Gedeon Richter Plc, Budapest, Hungary) via its dissolution in water and spraying into their feed. The goats in the CON group had their feed sprayed with the same amount of water. The dose of bromocriptine (0.06 mg/kg BW) used in this experiment was based on the studies by Dicks et al. [42] and Zhang et al. [38] and on the BCR instructions (about 0.05 to 0.07 mg/kg BW). The BCR treatment lasted for 30 days.

The feed intake of the goats was recorded daily during the experiment. The BW of the goats was recorded before morning feeding on days 0 and 30. Blood was collected from the goats using a 5 mL vacuum procoagulant tube before morning feeding on days 0, 15, and 30. The blood samples of the goats were immediately centrifuged at 1200× *g* for 10 min, and the resulting serum samples were stored at −20 °C. After blood collection on day 30, all goats were transported to the local slaughterhouse, and samples of their liver, subcutaneous adipose, and perirenal adipose tissues were collected immediately after slaughter. A portion of the liver, subcutaneous adipose tissue, and perirenal adipose tissue was immersed in 4% paraformaldehyde for subsequent histologic analysis. Another portion of their liver, subcutaneous adipose, and perirenal adipose tissues was collected and immediately frozen in liquid nitrogen for subsequent RNA extraction.

### 2.3. Serum Biochemical Index Analyses

The serum levels of triglyceride (TG, A110-1-1), total cholesterol (TCH, A111-1-1), low-density lipoprotein cholesterol (LDL-C, A113-1-1), high-density lipoprotein cholesterol (HDL-C, A112-1-1), fatty acid synthase (FAS, H231-1-2), 3-hydroxy-3-methylglutaryl-CoA reductase (HMGR, H236), acetyl-CoA carboxylase (ACC, H232-1-2), and hormone-sensitive lipase (HSL, H238-1-2) were analyzed using commercial assay kits provided by the Nanjing Jiancheng Bioengineering Institute (Nanjing, China, http://www.njjcbio.com/ accessed on 10 October 2020): TG, TCH, LDL-C, HDL-C, FAS, HSL, HMGR, and ACC Assay Kit Lot No. 20201010. Serum samples were obtained from 20 cashmere goats. TCH was measured using a COD-PAP enzymatic assay. TG was measured using a GPO-PAP enzymatic assay. HDL-C and LDL-C were measured using the dual reagent direct method. FAS, HSL, HMGR, and ACC were measured via an enzyme-linked immunosorbent assay (ELISA). The sensitivity of each assay was 0.2 ng/mL (FAS), 0.2 ng/mL (HSL), 0.2 ng/mL (HMGR), and 0.1 ng/mL (ACC). The intra-assay coefficients of variation were ≤ 8%. The fit indicators refer to R^2^ of the standard curves, which are all higher than 0.99. The analysis of each indicator was performed in duplicate.

### 2.4. Tissue Hematoxylin–Eosin Staining and Histology

The liver, subcutaneous adipose, and perirenal adipose tissues collected from the goats were fixed in 4% paraformaldehyde and stored overnight at room temperature. The fixed tissue was dehydrated using a fully automated dehydrator (JT-12S, Wuhan, China) in a gradient ethanol bath of increasing concentrations (75, 85, 95, and 100%). Paraffin was used to infiltrate the tissue. Each paraffin block was serially sectioned using a rotary slicer (Leica-2016, Wetzlar, Germany), and the slices were 5 μm thick. Xylene and ethanol were used for deparaffinizing the sections. Afterward, they were stained with HE, dehydrated, and sealed with neutral gum. Eight goats in each group were used for the HE staining experiments.

The Panoramic 250 digital section scanner from 3DHISTECH (Budapest, Hungary) was used to acquire images of the sections. General lesions on the tissues were observed at 40× and 200× magnification to observe areas of specific pathological changes. Measurements were performed with the measurement tool in the Case-Viewer image analysis software (2.3.0.99276), where more than 300 adipocytes were collected from each set of samples, and adipocyte areas were measured [36,43].

### 2.5. Quantitative Real-Time PCR

Total RNA was extracted from each sample using the TRIzol™ reagent (Invitrogen, Carlsbad, CA, USA). The TransScript One-Step gDNA Removal and cDNA Synthesis SuperMix reagent Kit (TransGen Biotech, Beijing, China) was used to reverse transcribe the extracted total RNA (1 μg) into cDNA. Eleven genes related to lipid metabolism were selected for the quantitative real-time PCR (qPCR) experiments: *PRL*; long-form PRL receptor (*LPRLR*); short-form PRL receptor (*SPRLR*); sterol regulatory element binding transcription factor 2 (*SREBF2*); sterol regulatory element binding transcription factor 1 (*SREBF1*); acetyl-CoA carboxylase alpha (*ACACA*); fatty acid synthase (*FASN*); 7-dehydrocholesterol reductase (*DHCR7*); 3-hydroxy-3-methylglutaryl-CoA reductase (*HMGCR*); peroxisome proliferator-activated receptor gamma (*PPARG*); and lipase E, hormone-sensitive type (*LIPE*). *β-actin* was used as a housekeeping gene for the relative qPCR experiments.

The qPCR was performed using ABI Step One PlusTM (Life Technologies, Carlsbad, CA, USA) and SYBR Green Supermix (Takara, Shiga, Japan). The amplification cycling conditions were as follows: 95 °C for 2 min (pre-denaturation), followed by 40 cycles of 95 °C for 5 s (denaturation) and 60 °C for 30 s (annealing and extension). Details of the primers used for qPCR are shown in Table 1. The specificity of each primer set was tested using a melting curve analysis for all genes in the range of 60 to 95 °C. Each qPCR analysis was performed in triplicate. All assays were run three times. The relative gene expression levels were calculated using the 2^−ΔΔCT^ method [44].

### 2.6. Statistical Analyses

The data were tested for independence, normal distribution, and homogeneity using SPSS 21.0 statistical analysis software. Data on CON and BCR groups were statistically analyzed using Student’s t-tests. Gene expression changes in the three tissues were analyzed using a one-way ANOVA, and differences between the groups were evaluated using Duncan’s multiple comparison test. The data were expressed as the mean ± standard error of the mean. *p* < 0.05 was considered statistically significant.

## 3. Results

### 3.1. Effects of PRL Inhibition on Body Weight and Feed Intake of Cashmere Goats

The BW and average daily feed intake (ADFI) of the cashmere goats in both groups are shown in Table 2. The effect of the average daily gain (ADG) and ADFI of cashmere goats that underwent BCR treatment during the experimental period was inconclusive (*p* > 0.05; Table 2).

### 3.2. Effects of PRL Inhibition on Serum Lipid Metabolism-Related Indexes

Our previous results showed that the inhibition of PRL decreased the serum PRL levels in goats [41]. Then, we tested the serum-related indexes of lipid metabolism and found that the HSL levels decreased in the BCR group on day 30 (*p* < 0.05; Figure 1F). PRL inhibition had an inconclusive effect on HSL levels at days 0 and 15 and inconclusive effects on the levels of TCH, TG, HDL-C, LDL-C, FAS, HMGR, and ACC (*p* > 0.05; Figure 1).

### 3.3. Effects of PRL Inhibition on the Histology of Liver, Subcutaneous Adipose, and Perirenal Adipose Tissues

The results of the histopathological and morphological analyses showed that the liver tissues of the two groups had intact peritonea, the lobules of the liver were not obviously lobulated, and the hepatic cords were relatively neatly arranged (Figure 2A,B). The endothelial cells of the central vein were relatively intact, and the hepatocytes were arranged radially around the central vein, with normal morphology and no obvious degeneration and necrosis. The structure of the hepatic sinusoids was normal, with no clear evidence of congestion, dilatation, or inflammatory infiltration. The structures of the interlobular arteries, veins, and bile ducts in the portal area were relatively intact. Therefore, HE staining showed that PRL inhibition had inconclusive effects on the pathological changes in the liver (*p* > 0.05).

HE staining of the adipose tissue also showed that PRL inhibition had inconclusive effects on the pathological changes in the subcutaneous (Figure 2C,D) and perirenal adipose tissues after PRL inhibition (*p* > 0.05; Figure 2E,F). Furthermore, adipocyte measurements showed an inconclusive effect of PRL inhibition on the area of subcutaneous adipocytes in both groups (*p* > 0.05; Figure 2G), while the area of perirenal adipocytes decreased in the BCR group (*p* < 0.05; Figure 2H).

### 3.4. Effects of PRL Inhibition on Lipid Metabolism-Related Genes in Liver, Subcutaneous Adipose, and Perirenal Adipose Tissues

The changes in the expression of 11 genes related to lipid metabolism in the liver, subcutaneous adipose, and perirenal adipose tissues after PRL inhibition are shown in Figure 3. In the liver, the expression of the *PRL*, *LPRLR*, *SPRLR*, *SREBF1*, *SREBF2*, *FASN*, *ACACA*, *HMGCR*, *DHCR7*, *PPARG*, and *LIPE* genes increased in the BCR group (*p* < 0.05; Figure 3A,B). In the subcutaneous adipose tissue, after the inhibition of PRL, the expression of the *SPRLR* gene increased (*p* < 0.05; Figure 3C) and the expression of the *SREBF1*, *SREBF2*, *ACACA*, *PPARG*, and *LIPE* genes decreased in the BCR group (*p* < 0.05; Figure 3C–E). In the perirenal adipose tissue, the expression of the *PRL*, *SREBF2*, and *HMGCR* genes decreased in the BCR group (*p* < 0.05; Figure 3F–H).

### 3.5. Changes in the Expression of PRL, LPRLR, SPRLR, and Lipid Metabolism-Related Genes in Liver, Subcutaneous Adipose, and Perirenal Adipose Tissues

We investigated the changes in the expression of different lipid metabolism genes in three tissues, using the liver as a benchmark. The results showed that the expression of the *PRL* gene was higher in subcutaneous adipose tissue than in the liver (*p* < 0.05). The *PRL* gene expression pattern was as follows: subcutaneous adipose tissue > perirenal adipose tissue > liver tissue (Figure 4D). Additionally, the gene expression pattern for *LPRLR* and *SPRLR* was as follows: liver tissue > subcutaneous adipose tissue > perirenal adipose tissue (Figure 4A). The expression of the *PRLR* gene was higher in the liver than in both types of adipose tissue (*p* < 0.05).

The *SREBF1* gene expression pattern was as follows: subcutaneous adipose tissue > liver tissue > perirenal adipose tissue (Figure 4B). The gene expression pattern of *SREBF2* was exactly the same as that of *PRLR*, being higher in the liver than in the subcutaneous and perirenal adipose tissues (*p* < 0.05; Figure 4B). The expression of the *DHCR7* gene was higher in the subcutaneous adipose tissue than in the perirenal adipose tissue (*p* < 0.05). The *DHCR7* gene expression pattern was as follows: subcutaneous adipose tissue > liver tissue > perirenal adipose tissue (Figure 4B).

The expression of the *FASN* and *ACACA* genes was higher in the subcutaneous adipose tissue than in the perirenal adipose and liver tissues (*p* < 0.05). The pattern of *FASN* and *ACACA* gene expression was as follows: subcutaneous adipose tissue > perirenal adipose tissue > liver tissue (Figure 4C). The expression of the *HMGCR* gene was higher in the subcutaneous adipose tissue than in the liver and perirenal adipose tissues (*p* < 0.05). The *HMGCR* gene expression pattern was as follows: subcutaneous adipose tissue > liver tissue > perirenal adipose tissue (Figure 4C). The expression of the *PPARG* and *LIPE* genes was higher in the subcutaneous adipose tissue than in the perirenal adipose tissue and higher in the perirenal adipose than in the liver tissue (*p* < 0.05; Figure 4D).

## 4. Discussion

### 4.1. Effects of PRL on Serum Lipid Metabolism Indexes

Traditionally, PRL has been known to play roles in lactation [2] and reproduction [45], but recent studies have highlighted an emerging role of PRL in metabolic homeostasis. PRL can directly affect metabolism by acting on the hypothalamus, pancreas, liver, and adipose tissues [4,12,34,46,47]. Additionally, the effect of PRL on an individual’s metabolism has been found to be closely related to its serum concentration [6]. Similarly, in our study, the serum concentration of PRL in cashmere goats significantly decreased after BCR treatment. Furthermore, a study on pigs found that BCR treatment had no effect on body weight and feed intake [48]. This finding is consistent with that in our study, where the BCR treatment had inconclusive effects on the body weight and average daily feed intake of cashmere goats. Previous studies also found that low and very high levels negatively affect metabolism, resulting in weight changes, dyslipidemia, and an increased prevalence of metabolic syndrome [8,9,12,13]. Another study showed that PRL inhibited lipid storage in adipose tissue [6]. PRL also ameliorated steatosis in the liver [34]. HSL is an important rate-limiting enzyme in lipolysis [49], and its reduction adversely affects the body’s lipid metabolism [50]. In our study, the serum level of HSL decreased after the inhibition of PRL. This suggests that low PRL levels reduce the rate of lipolysis in goats to some extent. In contrast, the inconclusive changes in serum TCH and TG levels, perhaps due to the short-term inhibition of PRL and the short duration of HSL reduction. In general, a low level of PRL affects the homeostasis of lipid metabolism in cashmere goats.

### 4.2. Effects of PRL on the Histology of the Liver, Subcutaneous Adipose, and Perirenal Adipose Tissues

PRL has recently been identified as a metabolic hormone, demonstrating the importance attached to its metabolic role [51]. Studies have shown that PRL can directly affect the metabolic activity of individuals by interacting with liver and adipose tissues. It has been found that PRL prevents hepatic steatosis in male mice on a high-fat diet and reduces TG accumulation in the liver of female mice [52]. In humans, PRL has also been shown to ameliorate hepatic steatosis via the CD36 pathway [34]. However, the role of PRL in lipid metabolism in the liver remains poorly understood. To investigate the effect of PRL on lipid metabolism in the liver, we observed the histomorphology of its sections. Through pathological observation of the sections, our results on the effects of PRL inhibition on pathological changes in the liver were inconclusive. Nevertheless, the relationship between PRL and lipid metabolism in the liver needs to be further studied.

With the discovery that PRL plays an important role in metabolism, PRL release from adipose tissue was observed in a study using human adipose tissue as a negative control for PRL release [25]. Biologically, subcutaneous and perirenal adipose tissues show significant differences in phenotypic architecture and lipid metabolism [36]. Compared with perirenal adipose tissue, subcutaneous adipose tissue has a stronger uptake of circulating free fatty acids and triglycerides and has a greater capacity for lipid deposition [53]. PRL release from human adipose depots has been found to be associated with adipose localization, differentiation status, and body mass index [54]. In our study, PRL inhibition showed inconclusive effects on the pathological changes in both types of adipose tissue. The subcutaneous adipocyte area did not change, while the perirenal adipocyte area decreased in the BCR group. The difference between subcutaneous and perirenal adipose tissues may be related to the difference in adipose localization. In patients who are obese, more PRL was released from the subcutaneous adipose tissue extracts than from the visceral adipose tissue extracts, and PRL release was less important in these patients than in patients who are lean [54]. A study showed that BCR reduced the amount of mesenteric and perirenal adipose tissue in lambs [55], which is consistent with our observation of a smaller area of perirenal adipocytes in the BCR group. However, bromocriptine had no effect on rib fat in lamb [56]. These findings indicate differences in the regulation of PRL on subcutaneous and perirenal adipose tissues, but it cannot be denied that PRL affects lipid metabolism homeostasis in adipose tissue.

### 4.3. Effects of PRL on the Expression of Genes Related to Lipid Metabolism

Previous studies on the relationship between PRL and lipid metabolism mainly focused on lactation or mammary tissue in animals [19,33,57]. During lactation in rats, PRL inhibited lipogenesis in adipose tissue by reducing FAS and ACC activity [57]. A study found that very high concentrations of PRL inhibited *FAS* expression in 3T3-L1 cells [33]. In goat mammary epithelial cells, the PRL treatment of cells increased the expression of the *PPARG*, *SREBF1*, *FASN*, and *ACACA* genes [19]. Lipid metabolism is a complex physiological process regulated by multiple genes and multiple pathways, and the genes involved in lipogenesis include *SREBF1*, *SREBF2*, *FASN*, *ACACA*, *HMGCR*, *DHCR7*, *PPARG*, etc. [58,59]; lipolysis is mainly regulated by HSL, and the gene encoding HSL is *LIPE* [60]. Since the liver is the center of metabolic homeostasis [4], we analyzed the changes in genes related to lipid metabolism in the liver. Our study found that the expression of the *PRL*, *LPRLR*, and *SPRLR* genes increased after the inhibition of PRL in the liver. At the same time, the expression of the *SREBF1*, *SREBF2*, *FASN*, *ACACA*, *HMGCR*, *DHCR7*, *LIPE*, and *PPARG* genes related to lipid metabolism increased after PRL inhibition in the liver. This finding is not consistent with the changes in the fat synthesis genes observed in goat mammary epithelial cells after PRL treatment [19], which may be related to the differences in liver and mammary tissue. PRLR has also been found to directly up-regulate SREBF1 or indirectly up-regulate SREBF1 via phosphorylating glycogen synthase kinase 3 [19,61]. In our study, genes involved in fat synthesis and lipolysis were simultaneously elevated in the liver. On the one hand, the reduction in the serum PRL levels may have affected the homeostasis of lipid metabolism in the liver. To maintain a homeostatic balance, the body increases lipid synthesis and lipolysis genes simultaneously, which may account for the inconclusive pathological changes in the liver tissue. On the other hand, when the body inhibits PRL secretion, PRL may affect both lipid synthesis and lipolysis in the liver through LPRLR and SPRLR. These results provide more insights into the potential metabolic role of PRL in the liver.

Previous studies have shown that visceral adipose has a higher lipid turnover than subcutaneous adipose [36]. In addition, visceral adipose has more androgen receptors, is sensitive to β-adrenergic hormones [62], and has greater lipolytic metabolic activity and insulin resistance [53]. The release of PRL is related to the localization of adipose [54], and the function of PRL is not consistent in different species. PRL has been found to have an anti-lipolysis effect on human subcutaneous abdominal explants, but this effect has not been observed in mouse adipose tissue [35]. In PRLR knockout mice, the development of subcutaneous adipose tissue was impaired [63]. Our study showed that in subcutaneous adipose tissue, PRL inhibition resulted in increased expression of the *SPRLR* gene and decreased expression of the *SREBF1*, *SREBF2*, *ACACA*, *PPARG*, and *LIPE* genes related to lipid metabolism in the BCR group. This may mean that lipid metabolic activity is suppressed in subcutaneous adipose tissue. The simultaneous decrease in lipogenic and lipolytic genes may account for the inconclusive effect of changes in the area of subcutaneous adipocytes.

In addition, LPRLR can induce an increase in fat storage, leading to hypertrophy of visceral fat depots [64]. The transduction pathways activated by LPRLR and SPRLR differ upon binding to PRL [38]. Interestingly, we found that only the expression of the *SPRLR* gene was significantly increased in subcutaneous adipose tissue, and the expression of both the *LPRLR* and *SPRLR* genes increased in the liver. At the same time, the changes in the lipid metabolism genes showed clear differences between the subcutaneous and perirenal adipose tissues, which may be caused by the different functions of *LPRLR* and *SPRLR*. Another study also showed that bromocriptine can reduce the amount of perirenal adipose tissue [55]. We observed a decrease in the *PRL* gene in the BCR group in perirenal adipose tissue, along with a decrease in *SREBF2* and *HMGCR* genes related to cholesterol synthesis. The decreased expression of the *SREBF2* and *HMGCR* genes may be responsible for the smaller area of perirenal adipocytes. Our study illustrates that the inhibition of PRL affected lipid metabolism in perirenal adipose tissue by decreasing the expression of the *PRL*, *SREBF2*, and *HMGCR* genes. In order to understand the metabolic role of PRL in different tissues better, we compared the changes in the expression of PRL-related genes in the three tissues. Previous studies have demonstrated that the expression of the *LPRLR* gene was higher in the liver of goats than that in their subcutaneous adipose tissue [19]; this finding was confirmed in our study. We also found that the expression of the *SPRLR* gene was higher in the liver than in adipose tissue, indicating the importance of PRLR in liver metabolism.

## 5. Conclusions

The inhibition of PRL had inconclusive effects on the body weight and average daily feed intake of cashmere goats but decreased their serum HSL levels. The effects of PRL on lipid metabolism were different in different tissues. PRL influences lipid metabolism by regulating different PRLRs in the liver and subcutaneous adipose tissue and by decreasing the expression of the *PRL*, *SREBF2*, and *HMGCR* genes in perirenal adipose tissue.

## Figures and Tables

**Figure 1 animals-14-03364-f001:**
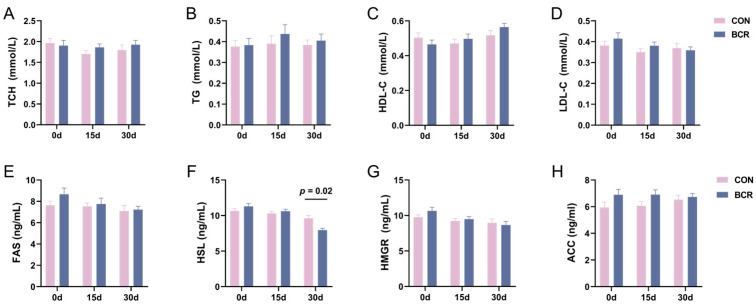
Changes in the serum lipid metabolism indexes in cashmere goats after PRL inhibition. (**A**) TCH, total cholesterol; (**B**) TG, triglyceride; (**C**) HDL-C, high-density lipoprotein cholesterol; (**D**) LDL-C, low-density lipoprotein cholesterol; (**E**) FAS, fatty acid synthase; (**F**) HSL, hormone-sensitive lipase; (**G**) HMGR, 3-hydroxy-3-methylglutaryl-CoA reductase; (**H**) ACC, acetyl-CoA carboxylase. CON, control group; BCR, bromocriptine treatment group; values are the mean ± standard error of the mean.

**Figure 2 animals-14-03364-f002:**
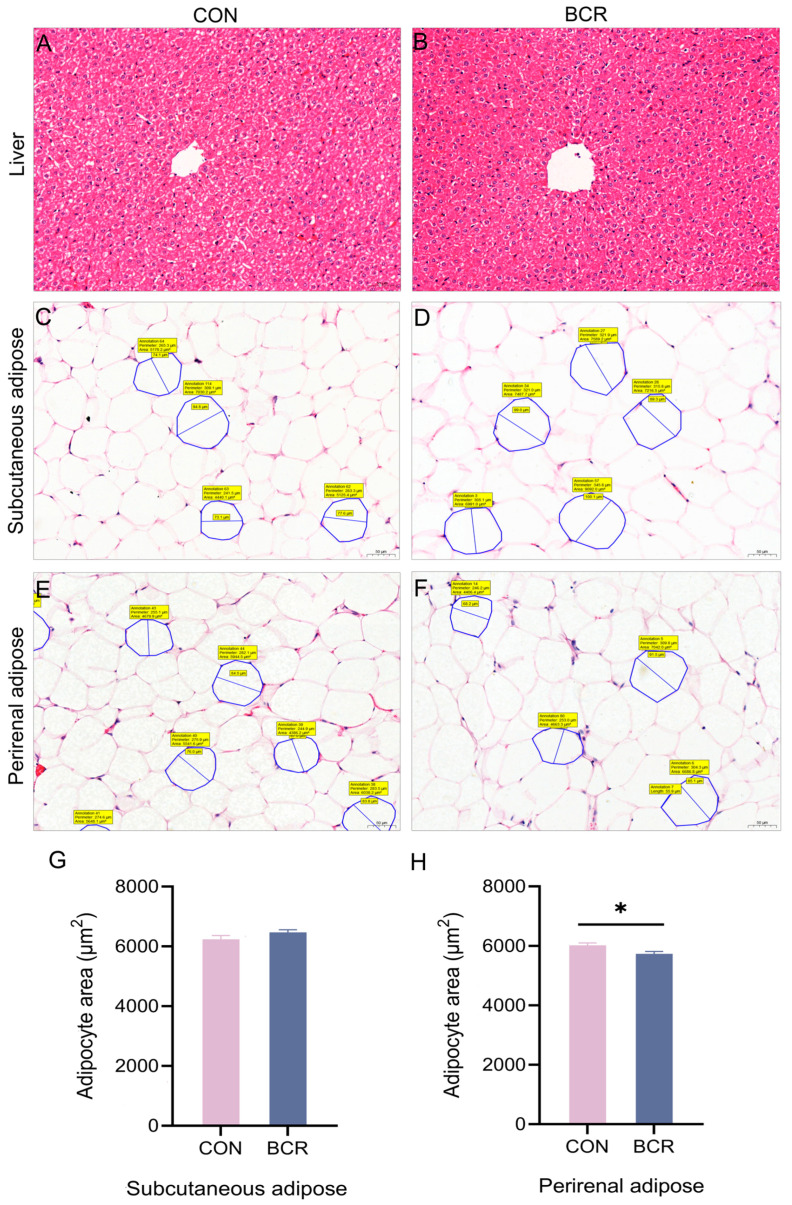
Morphometric changes in the liver, subcutaneous adipose, and perirenal adipose sections and measurement of adipocytes after PRL inhibition. (**A**,**B**) Liver sections from the CON and BCR groups; (**C**,**D**) subcutaneous adipose sections from the CON and BCR groups; (**E**,**F**) perirenal adipose sections from the CON and BCR groups; (**G**,**H**) changes in adipocyte area in the subcutaneous and perirenal adipose tissues; magnification × 200, scale bar = 50 μm. CON, control group; BCR, bromocriptine treatment group; values are the mean ± standard error of the mean. * *p* < 0.05.

**Figure 3 animals-14-03364-f003:**
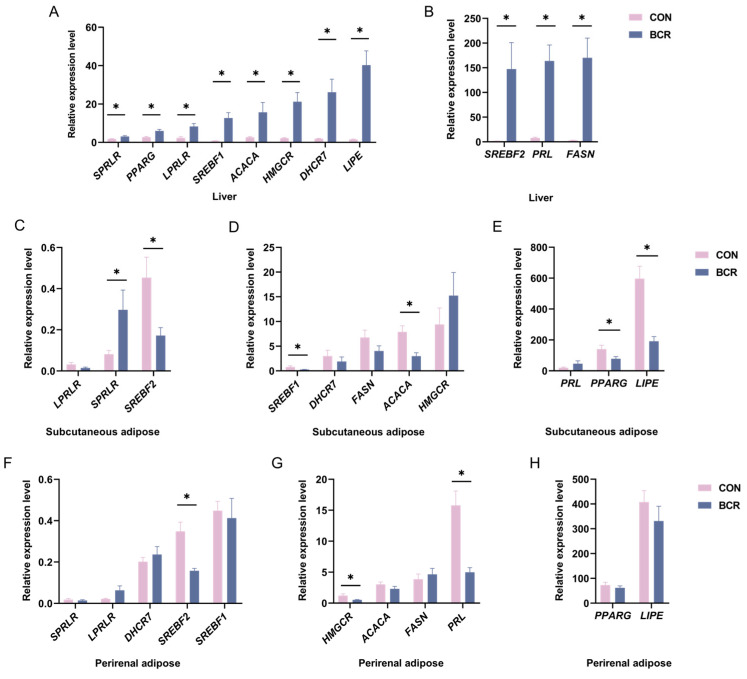
Dramatic changes in lipid metabolism-related genes in the liver, subcutaneous adipose, and perirenal adipose tissues. (**A**,**B**) Changes in lipid metabolism-related genes in the liver; (**C**–**E**) changes in lipid metabolism-related genes in subcutaneous adipose tissue; (**F**–**H**) changes in genes involved in lipid metabolism in perirenal adipose tissue. CON, control group; BCR, bromocriptine treatment group; values are the mean ± standard error of the mean. * *p* < 0.05.

**Figure 4 animals-14-03364-f004:**
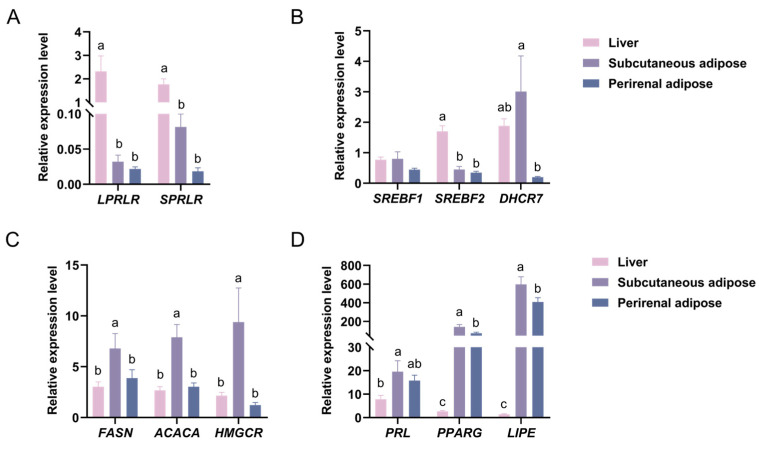
Changes in the expression of different lipid metabolism genes in three tissues. (**A**) Changes in the expression of *LPRLR* and *SPRLR* in three tissues. (**B**) Changes in the expression of *SREBF1, SREBF2*, and *DHCR7* in three tissues. (**C**) Changes in the expression of *FASN*, *ACACA*, and *HMGCR* in three tissues. (**D**) Changes in the expression of *PRL*, *PPARG*, and *LIPE* in three tissues. CON, control group; BCR, bromocriptine treatment group. Values are the mean ± standard error of the mean. ^a–c^ Different superscripts represented significant differences, *p* < 0.05.

**Table 1 animals-14-03364-t001:** Primer sequences.

Gene Name	Primer 5′ to 3′		Gene ID
*PRL*	F	TCCTGGAGCCAAAGAGACTG	100861193
	R	TGACGTGCCTCTTCATCCTT	
*LPRLR*	F	CTCAGGCCTATCCCTCCAAG	100861318
	R	TCGGGATTCTCCAGCTTCTC	
*SPRLR*	F	GCAGTGGCTTTGAAGGGCTAT	GU075814.1
	R	AGGCGAGAAGGCTGTGAT	
*SREBF1*	F	GCAAAGCCATCGACTACATC	100860908
	R	AGGTTCTCCTGCTTGAGCTT	
*SREBF2*	F	CATCATCGAGAAGCGGTATC	102189018
	R	CTCCTGGCGCAGTTTATGA	
*FASN*	F	CAGCCTCTTCCTGTTTGACG	100861286
	R	CATGAACTGCCGCATGAAGA	
*ACACA*	F	GCTGCCTCTGATGGTCTTTG	100861224
	R	GCCTGAGGAGGGATGTAGAC	
*HMGCR*	F	GGGAGCTTGCTGTGAGAATG	102169762
	R	TTCCATCCAAGCACAGAGGT	
*DHCR7*	F	CTGGACCCTCATCAACCTGT	102171848
	R	CGTGGCAGATGTCAATGGTT	
*PPARG*	F	CATTTCCGCTCCGCACTAC	100861309
	R	TGGAACCCTGACGCTTTATC	
*LIPE*	F	CAACGAGACTGGCATCAGTG	100860801
	R	GCACCTGGATCTCGGTGATA	
*β-actin*	F	GCGGCATTCACGAAACTACC	102179831
	R	GCCAGGGCAGTGATCTCTTT	

F, forward primer; R, reverse primer.

**Table 2 animals-14-03364-t002:** Effects of bromocriptine on the growth performance of cashmere goats.

Item ^1^	CON	BCR	*p* Value
IW, kg	23.86 ± 1.18	23.96 ± 1.20	*p* = 0.952
FW, kg	24.62 ± 1.26	24.58 ± 1.18	*p* = 0.986
ADG, g	25.19 ± 6.59	20.74 ± 7.79	*p* = 0.669
ADFI, kg	1.47 ± 0.02	1.46 ± 0.02	*p* = 0.659

^1^ IW, initial weight; FW, final weight; ADG, average daily gain; ADFI, average daily feed intake. CON, control group; BCR, bromocriptine treatment group. Values are the mean ± standard error of the mean. The same superscript represents no statistical difference.

## Data Availability

All datasets generated and analyzed during the current study are available from the corresponding author (zhangyingjie66@126.com and liuyueqin66@126.com) upon reasonable request.

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
