# Peer review of "Effects of Prolactin Inhibition on Lipid Metabolism in Goats"

_animals, 2024, doi:10.3390/ani14233364_

Round 1

Reviewer 1 Report

Comments and Suggestions for Authors

The authors of this study investigated the effects of the prolactin inhibitor bromocriptine on lipid metabolism in the liver, subcutaneous adipose tissue, and perirenal adipose tissue of cashmere goats.

There are some problems in the article, and the suggestions are as follows:

1. Line 192, Should be “forward primer”; “reverse primer”.

2. Line 202, Should be “Effects of PRL inhibition in body weight and feed intake of cashmere goats”

3. In result 3.2, What does the lower HSL levels in the BCR group at day 30 indicate? I think this result lacks a conclusion.

4. Add grouping information and tissue type in Figure 2 A-F.

5. In result 3.3, Explain why bromocriptine affected perirenal adipocyte size but did not alter subcutaneous adipocyte size?

6. In Figure 3 and 4, Gene names should be italicized.

Comments on the Quality of English Language

There are some verbal errors throughout the manuscript.

Author Response

The authors of this study investigated the effects of the prolactin inhibitor bromocriptine on lipid metabolism in the liver, subcutaneous adipose tissue, and perirenal adipose tissue of cashmere goats.

There are some problems in the article, and the suggestions are as follows:

  1. Line 192, Should be “forward primer”; “reverse primer”. 1.

Thanks for your kind suggestion, we have revised the manuscript (Lines 204).

  1. Line 202, Should be “Effects of PRL inhibition in body weight and feed intake of cashmere goats”

According to your suggestion, we have revised the manuscript (Lines 213).

  1. In result 3.2, What does the lower HSL levels in the BCR group at day 30 indicate? I think this result lacks a conclusion.

Based on your suggestion, we supplemented the corresponding information (Lines 322-327).

“Another study showed that PRL inhibited lipid storage in adipose tissue [6]. PRL also ameliorated steatosis in the liver [34]. HSL is an important rate-limiting enzyme in lipolysis [49], and its reduction adversely affects the body's lipid metabolism [50]. In our study, the serum level of HSL decreased after the inhibition of PRL. This suggests that low PRL levels reduce the rate of lipolysis in goats to some extent.”

  1. Add grouping information and tissue type in Figure 2 A-F.

According to your suggestion, we have revised the manuscript (Lines 253).

  1. In result 3.3, Explain why bromocriptine affected perirenal adipocyte size but did not alter subcutaneous adipocyte size?

Thanks for your kind suggestions, we supplemented the corresponding information (Lines 352-359).

“The difference between subcutaneous and perirenal adipose tissues may be related to the difference in adipose localization. In patients who are obese, more PRL was released from the subcutaneous adipose tissue extracts than from the visceral adipose tissue extracts, and PRL release was less important in these patients than in patients who are lean [54]. A study showed that BCR reduced the amount of mesenteric and perirenal adipose tissue in lambs [55], which is consistent with our observation of a smaller area of perirenal adipocytes in the BCR group. However, bromocriptine had no effect on rib fat in lamb [56].”

  1. In Figure 3 and 4, Gene names should be italicized.

According to your suggestion, we have revised the manuscript (Lines 272, 303).

Quality of English Language

(x) The English could be improved to more clearly express the research.

Thank you again for your suggestions, we have improved the Quality of English Language for the manuscript by editing service website (https://www.mdpi.com/authors/english- english-87424). 

All revisions in the manuscript have been highlighted.

Reviewer 2 Report

Comments and Suggestions for Authors

The article by Liu et al. explored the effects of prolactin inhibition on lipid metabolism in goats. The study fed bromocriptine, a prolactin inhibitor, to cashmere goats for 30 days and found that prolactin inhibition reduced serum hormone-sensitive lipase levels, however, the effects on lipid metabolism varied in different tissues. The manuscript is interesting and well written. Please consider the following feedback to further enhance its quality.

Simple Summary:

1. Line 36 Consider changing "feed intake" to " average daily feed intake".

Abstract:

3. Authors are advised to provide acronyms for assay parameters, such as body weight, average daily feed intake, serum parameters, and genes.

4. Why the author chose subcutaneous adipose tissue and perirenal adipose tissue, please give a brief explanation in the introduction.

5. Why bromocriptine was chosen as a prolactin inhibitor, and whether feeding bromocriptine would have other effects on cashmere goats besides reducing prolactin levels? It is suggested that the author clarify in the introduction.

6. Goats in the same group were fed individually or in herds? How do you ensure that each goat receives the same dose of bromocriptine if they were fed in herds?

7. Line164 It is suggested that the author revise the manuscript sentence as The general lesions of tissue were observed at 40× magnification images and 200× magnification images were acquired to observe areas of specific pathological changes.

8. Line 166-167 Delete this line.

9. Line 303, Line 348 and L393 I suggested that the author revise “Studies” to “A study”.

10. Line 346-347 The authors were advised to supplement references 19,51,52.

11. Line 372-373 The authors were advised to supplement references.

Author Response

The article by Liu et al. explored the effects of prolactin inhibition on lipid metabolism in goats. The study fed bromocriptine, a prolactin inhibitor, to cashmere goats for 30 days and found that prolactin inhibition reduced serum hormone-sensitive lipase levels, however, the effects on lipid metabolism varied in different tissues. The manuscript is interesting and well written. Please consider the following feedback to further enhance its quality.

Simple Summary:

  1. Line 36 Consider changing "feed intake" to " average daily feed intake".

Thanks for your kind suggestion, we have revised the manuscript (Line 36).

Abstract:

  1. Authors are advised to provide acronyms for assay parameters, such as body weight, average daily feed intake, serum parameters, and genes.

According to your suggestion, we have revised the manuscript (Lines 36-40).

  1. Why the author chose subcutaneous adipose tissue and perirenal adipose tissue, please give a brief explanation in the introduction.

Thank you for your kind suggestion, we supplemented the corresponding information (Lines 101-105).

“Adipose tissue is generally categorized into subcutaneous and visceral adipose tissues, and the two types of adipose tissue have distinct differences, with visceral adipose tissue possessing a higher lipid turnover rate than subcutaneous adipose tissue [36]. The simultaneous collection of two different types of adipose tissue is more conducive to exploring the role of PRL in lipid metabolism.”

  1. Why bromocriptine was chosen as a prolactin inhibitor, and whether feeding bromocriptine would have other effects on cashmere goats besides reducing prolactin levels? It is suggested that the author clarify in the introduction.

Thanks for your kind suggestion, we supplemented the corresponding information (Lines 105-107).

“Additionally, bromocriptine (BCR), a dopamine agonist that inhibits PRL secretion, has been widely used to study the function of PRL [37-40].”

  1. Goats in the same group were fed individually or in herds? How do you ensure that each goat receives the same dose of bromocriptine if they were fed in herds?

Thank you for your kind suggestion, we supplemented the corresponding information (Line 122).

“Twenty goats were housed in twenty individual pens.”

  1. Line164 It is suggested that the author revise the manuscript sentence as “The general lesions of tissue were observed at 40× magnification images and 200× magnification images were acquired to observe areas of specific pathological changes”. 7.

According to your suggestion, we have revised the manuscript (Lines 177-178).

  1. Line 166-167 Delete this line.

According to your suggestion, we have revised the manuscript.

  1. Line 303, Line 348 and L393 I suggested that the author revise “Studies” to “A study”.

According to your suggestion, we have revised the manuscript (Lines 317, 365, 410).

  1. Line 346-347 The authors were advised to supplement references 19,51,52.

Thanks for your suggestion, we have provided the statement of reference (Lines 363-364).

  1. Line 372-373 The authors were advised to supplement references.

Thanks for your suggestion, we have added the statement of reference (Lines 389-390).

Thank you again for your suggestions that helped us improve the quality of the manuscript.

All revisions in the manuscript have been highlighted.

Reviewer 3 Report

Comments and Suggestions for Authors

The authors have presented an original study evaluating the metabolic role of prolactin on lipid synthesis. Well written article and only minor corrections have been recommended. I have strong reservations about a protein concentrate with 30% crude ash content untill authors are able to prsente the composition.

Other specific comments detailed below:

L92/93: In obese women, PRL promotes  visceral fat storage[32]. Moreover, PRL inhibits lipid storage and adipokine release in adipose tissue[6]…. Authors should provide some contxt to the varyng role of PRL on fat storage.

L115: crude ash 30%? Authors should provide the composition of the protein concentrate.Why is the concentraton of some nutrients variable. calcium 1.5%-4.5%, Sodium chloride 1.0%-4.0%.

Table 2: Authors should provide the actual p-values that shows signicicance or no significance.

Figure 1F. Instead of a star sign, simply add the actual p-value.

L339/340: Re-write this sentence “It has been shown that BCR has been shown..”

L386: LPRLR can induce…

Author Response

The authors have presented an original study evaluating the metabolic role of prolactin on lipid synthesis. Well written article and only minor corrections have been recommended. I have strong reservations about a protein concentrate with 30% crude ash content until authors are able to prsente the composition.

Other specific comments detailed below:

L92/93: In obese women, PRL promotes visceral fat storage[32]. Moreover, PRL inhibits lipid storage and adipokine release in adipose tissue[6]    Authors should provide some context to the varyng role of PRL on fat storage. 1992 /93:

Thank you for your suggestion, we supplemented the corresponding information (Lines 95-99).

“PRL can inhibit the mRNA and protein expression of fatty acid synthase (FAS) in adipocytes [33]. In the liver, PRL ameliorates hepatic steatosis via the CD36 pathway [34]. PRL has been shown to inhibit isoproterenol-stimulated lipolysis in rat and human adipose tissues [27] but has no effect on lipolysis in mouse adipose explants [35].”

L115: crude ash 30%? Authors should provide the composition of the protein concentrate. Why is the concentration of some nutrients variable. calcium 1.5%-4.5%, Sodium chloride 1.0%-4.0%.

Thanks for your kind suggestion, the nutritional composition information in the manuscript is described according to the protein concentrate feed instructions, which mainly provide crude protein content of ≥30% , crude ash content of ≤30%, calcium 1.5%-4.5%, and sodium chloride 1.0%-4.0%. We contacted the manufacturer and retested the protein concentrate to obtain more specific nutritional information.

The protein concentrate included crude protein 43.8%, crude ash 24.7%, crude fiber 12%, calcium 1.9%, sodium chloride 2.4%, lysine 1.2%, total phosphorus 1.0%, and moisture 13.0% (Lines 126-128).

Table 2: Authors should provide the actual p-values that shows significance or no significance.

Thank you for your suggestion, we have revised the manuscript (Line 218).

Figure 1F. Instead of a star sign, simply add the actual p-value.

According to your suggestion, we have revised the manuscript (Line 228).

L339/340: Re-write this sentence “It has been shown that BCR has been shown..”

Thank you for your suggestion, we have revised the manuscript (Lines 356-357).

L386: LPRLR can induce…

According to your suggestion, we have revised the manuscript (Line 403).

Thank you again for your suggestions that helped us improve the quality of the manuscript.

All revisions in the manuscript have been highlighted.